# Taxonomy of the Genus *Halophila* Thouars (Hydocharitaceae): A Review

**DOI:** 10.3390/plants9121732

**Published:** 2020-12-08

**Authors:** John Kuo

**Affiliations:** Centre for Microscopy, Characterisation and Analysis, The University of Western Australia, 35 Stirling Highway, Perth 6009, WA, Australia; john,kuo@uwa.edu.au or john.js.kuo@gmail.com

**Keywords:** *Halophila* taxonomy, history, sections, typifications, combinations, keys, descriptions, illustrations

## Abstract

The seagrass genus *Halophila* Thouars has more than twenty described species and is predominately distributed over a wide geographic range along the tropical and the warm temperate coastlines in the Indo-West Pacific Oceans. A brief history of the *Halophila* taxonomic development is presented. Based on reproductive and vegetative morphology, the genus is divided into eight sections including three new sections: section Australes, section Stipulaceae and section Decipientes. A rewritten taxonomic description of the type species for the genus *Halophila,*
*H. madagascariensis* Steudel ex Doty et B.C. Stone, is provided. The lectotype of *H. engelmannii* Asch. as well as neotypes of *H. hawaiiana* Doty et B.C. Stone and *H. spinulosa* (Br.) Asch. are designated. Furthermore, *H. ovalis* ssp. *bullosa*, ssp. *ramamurthiana* and ssp. *linearis* together with *H. balforurii* have been recognised as distinct species. Nomenclature, typification, morphological description and botanical illustrations are presented for each taxon. Recent molecular phylogenetic surveys on certain *Halophila* taxa are also discussed. Field surveys for the deep water *Halophila* in West Pacific regions are suggested. Morphological studies combined with molecular investigations for the *Halophila* on the east coast of Africa and the West Indian Ocean are urgently needed and highly recommended.

## 1. Introduction

The seagrass *Halophila* is the smallest in size among the entire known seagrasses, but it is the most diverse group and represents more than one quarter of all recognised seagrass species [1]. *Halophila* plants are minute and fragile, without stripe-like long leaf blades, and virtually unlike any other seagrasses. As a result, their taxonomic position in the monocotyledonous classification has long been unsettled. Ascherson [2] placed *Halophila* under the tribe Halophileae in the family Najadaceae and then Ascherson and Gürke assigned it as the tribe Halophiloideae within the family Hydrocharitaceae [3]. Rydberg [4] later included *Halophila* in the family Elodeaceae instead of the family Hydrocharitaceae. Den Hartog [5] considered *Halophila* as one of nine genera in the Hydrocharitaceae; however, in his monograph [6], he followed Nikai’s interpretation [7] and placed *Halophila* under the subfamily Halophiloideae (one genus only) in the family Hydrocharitaceae. Den Hartog and Kuo [1] stressed that the subfamily Halophiloideae status should be maintained for the *Halophila*.

Ascherson [8] divided the *Halophila* into two sections: Barkina (containing *H. ovalis*) and Microhalophila (consisting of *H. beccarii*). However, none of his subsequent publications used the sections in the genus *Halophila*. Ostenfeld [9] divided *Halophila* (8 species) into four sections: the section Typicae—*H. ovalis*, *H. stipulacea*, *H. baillonis* and *H. decipiens*; the section Americanae—*H. engelmannii* and *H. aschersonii*; the section Spinulosae—*H. spinulosa*; and the section Pusillae—*H. beccarii.* The first three sections possess stipules and the fourth section Pusillae has none. Ostenfeld and Meeresgraser [10] made minor alterations in the species compositions for these four sections. The section Typicae included *H. ovata* to replace *H. stipulacea*, which was in turn transferred to the section Spinulosae. Den Hartog [5] proposed the section Aschersonae for *H. spinulosa* and re-elected the section Microhalophila of Ascherson to replace the section Pusillae, as well as comprised the section Halophila to substitute the section Typicae. He later [6] discarded the section Aschersonae and re-installed the section Spinulosae. Greenway [11] created a new section Tricostatae that was based on the species *H. tricostata* from the Great Barrier Reef, Australia. Thus, the genus *Halophila* has become five sections. 

The separation of the above-mentioned sections in *Halophila* was established mainly by relying on vegetative morphological characteristics and did not take generative characteristics into account, probably due to the lack of sufficient reproductive information. As a result, up to the present, all *Halophila* species with short erect lateral shoots have been considered to belong to the section Halophila. Therefore, the section Halophila contains both monoecious (e.g., *H. decipiens; H. capricorni*) and dioecious (e.g., *H. ovalis, H. stipulacea*) species. However, based on both vegetative and generative characteristics, this review has proposed three new sections in the genus *Halophila*: section Australes, section Stipulaceae and section Decipientes.

Prior to the establishment of genus *Halophila* by Thouars [12], the three currently recognised *Halophila* species (*H. ovalis*, *H. stipulacea*, *H*. *spinulosa*) had already been described under other genera as *Zostera stipulacea* [13], *Caulinia ovalis* and *C. spinulosa* [14]. Gaudichard [15] called *H. ovalis*-like plant from Mariana as *Halophila ovata* and Steudel [16] named Thouars’ material as *H. madagascariensis*. In 1854, Zollinger described two new seagrass species from Indonesia as *Lemonips major* and *L. minor* [17]. Subsequently, Miquel [18] transferred the former species only to *Halophila*, which became *H. major*, and named the latter species as *Halophila lemonips*. Ascherson [2] initially recognised only two species of *Halophila*, *H. stipulacea* and *H. ovalis*, and under his interpretation, *H. ovalis* contained three varieties, *H. minor*, *H. major* and *H. ovata*, and he treated *H. madagascariensis* as a synonym of *H ovalis*. However, by 1871, he abandoned these varieties and presumably included them within *H. ovalis* [19]. Ascherson also described three new species as *H. beccarii* [8], *H. baillonis* [20] and *H. engelmannii* [21]. By 1875, Ascherson [21] recognised six species of *Halophila* including *H. stipulacea*, *H. ovalis* (without varieties), *H. spinulosa*, *H. beccarii*, *H*. *baillonis* and *H. engelmannii*. Ascherson [22] included eight species in the genus *Halophila* by adding two newly described species, *H. decipiens* and *H. aschersonii*, from Thailand and Virgin Islands, respectively, by Ostenfeld [9,23]. Ostenfeld [24] re-elected *H. ovata* from *H. ovalis* to be a distinct species and consequently [10] he recognised nine species in the genus *Halophila*. Makino [25] described a new species, *H euphlebia*, from the Japanese waters. When Solereder [26] noticed that *H. stipulacea* of Rodriguez differed from those from Egypt, he renamed the material from the former locality as *H. balfourii*. Setchell [27] described a new variety of *H. ovalis* as *H. ovalis* var. *bullosa* from the American Samoa. In 1957, Den Hartog [5] also identified eight *Halophila* species but included the variety of *H. bullosa* and omitted *H. aschersonii,* which he considered as a synonym of *H. baillonis*. In addition to Setchell’s [28] list, den Hartog added a new variety in *H. decipiens* as *H. decipiens* var. *pubescens* and transferred *Lemonips minor* Zollinger into the genus *Halophila* [5]. In addition, he described a new species, *H. linearis*, from Mozambique [29]. Doty and Stone [30] described two new species, *Halophila hawaiiana* and *H. australis*, which were based on the material from Hawaii and South-Eastern Australia, respectively. Doty and Stone [31] also contributed a formal species description for *H. madagascariensis* from Thouars’ material and designated it as the lectotype of the genus *Halophila*. It should be noted that earlier in 1946, Bathie described and illustrated the species (the same Thouars’ material) under the name of *H. ovata* [32].

Den Hartog [6] reclassified *Halophila* into eight species and considered *H. balfourii*, *H. madagascariensis* and *H. decipiens* var. *pubescens* as synonyms of *H. stipulacea, H. ovalis* and *H. decipiens*, respectively, and he also used the name *H. ovata* to replace *H. minor*. Den Hartog [6] further considered *H. ovalis* as a “collective species” and treated *H. bullosa*, *H. hawaiiana, H. australis* and *H. linearis* as four subspecies of *H. ovalis*. Sachet and Fosberg [33] cited morphological reasons for treating ssp. *H*. *hawaiiana* as a distinct species but closely related to *H. minor* and transferred *H. ovalis* ssp. *bullosa* into *H. minor* ssp. *bullosa*.

Since den Hartog’s [6] monograph, there have been further developments in *Halophila* taxonomy. These include *H. tricostata* from the Great Barrier Reef [11]; *H. johnsonii* from Florida [34]; *H. ovalis* ssp. *ramamurthiana* from South-Eastern India [35] and *H. capricorni* from the Coral Sea [36]. Furthermore, Robertson [37] restated *H. australis* as a distinct species and Kuo [38] demonstrated that *H. minor* and *H. ovata* were two distinct species. Kuo and den Hartog [39] provided keys and brief descriptions for 14 species and three subspecies of *H. ovalis* in five sections of the genus *Halophil*a. In addition, Den Hartog and Kuo [1] listed five sections and fifteen *Halophila* species with four subspecies of *H. ovalis* and recognised *H. madagascariensis* as the type species of the genus *Halophila*. Furthermore, Kuo et al. [40] applied light microscopic techniques to revise the Japanese *Halophila* species and described three new species as *H. mikii, H*. *nipponica* and *H. okinawensis* and treated *H. euphlebia* as a synonym of *H. major*. In addition, it was proposed to use the new name of *H. gaudichardii* to replace *H. ovata*, based on nomenclature illegitimacy. Later in that year, Uchimura et al. [41] used the molecular data generated from the material collected from Japan to describe a new species as *Halophila japonica*. Most recently, a new monoecious species *H. sulawesii* was added, which was considered as “deep-water *H. ovalis*” from Sulawesi Islands, Indonesia [42]. Currently, *The Plant List* recognises 20 *Halophila* species names. The *Halophila* classification history reported here is summarised in Table 1.

Despite Doty and Stone [31] designating *H. madagascariensis* as lectotype for the genus *Halophila*, den Hartog [6] treated type species as *H. madagascariensis* Steudel (= *H. ovalis* (R. Br.) Hook. ƒ.) and listed *H. madagascariensis* as a synonym of *H. ovalis*. Den Hartog and Kuo [1] finally listed *H. madagascariensis* (Steudel) Doty & B.C. Stone as the type species of the genus *Halophila*. A complete re-description of this type species is provided in this review. Although the type locality has been identified as Florida [4], the typification for *H. engelmannii* has not been accomplished. Furthermore, the holotypes of *H. spinulosa* at BM and *H. hawaiiana* at BISH have been lost. Therefore, the designation of neotypes for these species is required and will be designated in this review.

Waycott et al. [43] conducted a phylogenetic relationship investigation in the genus *Halophila* using 11 recognised Halophila species: *H. ovalis*, *H. beccarii*, *H. engelmannii*, *H. tricostata*, *H. spinulosa*, *H. decipiens*, *H. stipulacea*, *H. hawaiiana*, *H. johnsonii*, *H. minor* and *H. australis*. They found that most of their studied species appeared to be distinct and were associated with vegetative morphological differences. However, both *H. johnsonii* and *H. hawaiiana* were not separated from *H. ovalis*, and their Southern Australian endemic species *H. australis* was closely related to some Halophila populations in North-Eastern Australia and also in the Philippines. On the other hand, Japanese molecular scientists [44] found that only *H. ovalis*, *H. nipponica*, *H. major* and *H. decipiens* could be recognised molecularly, but they had treated other Halophila species: *H. minor*, *H. mikii*, *H. gaudichardii* and *H. okinawensis* from Japanese water as H. ovalis. Uchimura et al. [44] formally identified molecularly *H. australis* from the species type locality and indicated this *H. australis* by Waycott et al. [43] as H. major. In fact, the mis-identified *H. australis* could be *H. ovalis*. It was very surprising that Shimada et al. [45] identified molecularly that the Halophila specimens from Okinawa Islands deep water with the blade 20:1 (length:width) ratio as *H. nipponica* and not as *H. okinawensis*. Unfortunately, the blade length:width ratio of *H. nipponica* is only 2–3:1. These authors speculated that only *H. decipiens*, *H. ovalis* (including *H. hawaiiana*, *H. johnsonii*, *H. minor*), *H. major* (including *H. mikii*, *H. australis* by Waycott et. al. [43]) and *H. nipponica* (*H. gaudichardii* and *H. okinawensis*) were belonging to so-called *Halophila ovalis*–*H. minor* species complex. On the contrary, it was later demonstrated molecularly that *H. okinawensis* and *H. gaudichardii* were distinct taxa [46]. A series of molecular analyses on the Halophila species from South-East Asian countries (i.e., Vietnam, Myanmar, Thailand, Malaysia, India) and the Red Sea by Nguyen and Papenbock’s team [47,48,49,50,51] concluded that Halophila ovalis, *H. major*, *H. minor* (they referred as *H. ovata*), *H. decipiens* and *H. beccarii* were distinct species and *H. ovalis* ssp. ramamurthiana was genetically different from *H. ovalis* in South-Eastern India. Furthermore, they found that *H. beccarii* was closer to the ancestor of Halophila species than *H. ovalis* or *H. decipiens*, as indicated previously by Waycott et al. [43]. Nguyen et al. [50] suggested it that the Thai–Malay Peninsula was a geographic barrier between *H. ovalis* populations in the Western Pacific and the Eastern Indian Ocean is worthwhile to mention that all Nguyen and Papenbock’s molecular studies were accompanied by morphological data on the studied species.

Kuo and den Hartog [39] provided a key and brief description for all seagrasses globally including fifteen *Halophila* species. Some field guides of seagrasses including *Halophila* have also been compiled for the local regions, e.g., [52,53,54,55,56,57,58]. However, updated comprehensive taxonomic treatment and the key for the identification of the *Halophila* species globally have not been published since 2001.

For the flowering plants, the reproductive structures such as petals, sepals, stamens, fruits and seeds are main sources for species identification. However, flowers and fruits of most *Halophila* are not often collected. Only a small fraction of 2000 *Halophila* specimens studied from forty major herbaria (Appendix A) had reproductive material. Therefore, the identification of *Halophila* species and sections is more or less dependent on vegetative characteristics, such as plant appearance, blade length, width, blade tips, cross vein numbers, branching, etc. Sometimes, some of these vegetative characteristics may show considerable variation or overlapping. Nevertheless, if used in conjunction with other features such as geographic locations and habitats, the vegetative characteristics are of great value taxonomically. Therefore, vegetative characteristics and reproductive structures when available will be used for *Halophila* section classification, and vegetative characteristics alone will be employed in the key presented in this paper for species identification. The following key is designed and arranged to identify all described (known) *Halophila* sections and then the species. Under each taxon, the type locality, synonyms and morphological descriptions are given. The locality, collector and herbarium information of botanical drawings for the *Halophila* species are listed in Appendix A while Botanical drawings for each species will be placed in Appendix A.

## 2. Halophila Classification


**Genus Halophila Thouars**
**Halophila** Thouars, *Gen. Nov. Madagsc*. 2. (1806).Type species: *Halophila madagascariensis* Steud. ex Doty *et* B.C. Stone.*Barkania* Ehrenb., *Abh. Königl. Akad. Wiss. Berlin*, 1832, *I*, 429 (1834). T: not designated.*Lemnopsis* Zoll., *Syst. Verz*. 1854, *1*, 74–75, *nom. illeg*. *non* Zipp. (1829). T: not designated.

Marine to estuarine, submerged, sometimes intertidal, monoecious or dioecious, perennial, rare annual herbs. Rhizomes creeping, with 1 lateral shoot and 1 unbranched root at each node. Scales 2, one covering root base, the other covering lateral shoot base. Leaves in pair or in pseudo-whorl, sessile or petiolate; blades linear, lanceolate to ovate, entire or serrulate, glabrous or hairy, with a mid-vein and 2 intramarginal veins connected by cross veins; stomata absent; tannin cells absent. Spathes with 2 imbricate, keeled bracts; 1 flower or with a male and a female flower. Male flower pedicellate; tepals 3; stamens 3; anthers 2, oblong, sessile, dehiscing longitudinally; pollen grains ellipsoid, in long chains. Female flower sessile; hypanthium persistent, bearing 3 reduced tepals; ovary ovoid to ellipsoid, unilocular; styles 3–6, filiform. Fruits ellipsoid to globose, beaked. Seeds numerous, globose, testa mostly reticulate. 

The genus contains at least 24 described species, arranged within eight sections; mostly distributed in tropical and warm temperate waters. Some species are well defined and restricted in distribution, but a few species, such as *H. ovalis*, *H. major* and *H. minor*, are widely sympatric, distributed in the Indo-West Pacific region. *Halophila decipiens* is the only truly pantropic species which occurs in the Indian Ocean and tropical region of Pacific and W Atlantic Oceans, recently also in the Mediterranean Ocean. The *Halophila* has not been found in east coasts of both the Atlantic and the Pacific Oceans.


**Key to sections**
**1.** Erect lateral shoots (erect stems) extremely short or not distinct, bearing a pair of petiolate leaves; flowers mostly on rhizome nodes
**2**
**1:** Erect lateral shoots (erected stems) elongated, distinct, bearing leaves in pseudo-whorls or in many pairs of sessile leaves; flowers on erect later shoot nodes
**5**
**2.** Female flowers and fruits on extended floral shoots; styles 6
**Sect. 1. Australes**
**2:** Female flowers and fruits on rhizome nodes; styles 3
**3**
**3.** Leaf blade margin entire; surface glabrous
**Sect. 2. Halophila**
**3:** Leaf margin serrulate; surface hairy or spines, rarely glabrous
**4**
**4.** Dioecious; leaf blade distinctly linear; petiole sheathing lopsidedly
**Sect. 3. Stipulaceae**
**4:** Monoecious; leaf blade ovate to elliptic; petioles not sheathing
**Sect. 4. Decipientes**
**5.** Monoecious; minute plants having lateral shoots with 1–2 nodes, bearing a pseud-whorl of 6–10 leaves at the top; leaf blade without cross veins
**Sect. 5. Microhalophila**
**5:** Dioecious; lateral shoots with numerous leaf blade-bearing nodes or 2 nodes bearing a pseudo-whorl of leaves at top; leaf blade with cross veins
**6**
**6.** Lateral erect shoots with 2 scales at the base; having numerous leaf blade-bearing nodes or a pseudo-whorl of leaves at each node
**7**
**6:** Lateral erect shoots with 2 scales at the base and 2 other scales around halfway up; bearing a pseudo-whorl of leaves at top
**Sect. 6. Americanae**
**7.** Lateral erect shoots stiff, wiry; leaves in 10–20 pairs distichously arranged on each lateral shoot; leaf blade with basal folding
**Sect. 7. Spinulosae**
**7:** Lateral erect shoots soft; leaves in 2 or 3 pseudo-whorls at each node on lateral shoot; blade not basal folding
**Sect. 8. Tricostatae**



**Sect. 1. Australes**
*Halophil*a sect. *Australes* J. KuoType species: *Halophila australis* Doty & B.C. Stone.Dioecious. Male flowers form on the rhizome node; female flowers form on an erect extended lateral shoot. Leaf blade margin entire, surface glabrous.Dioca. Flores masculi ad geniculum rhizomatis facti sed feminei in surculum erectum productum laterale facti. Foliorum lamina integra, glabra.One species endemic to the temperate Australian south coast including Tasmania.

**Halophila australis** Doty & B.C. Stone, *Brittonia*, 1966, *18*, 306. (Appendix A).T: Australia Victoria, Queencliff, Jan. 1922, *Lucas s.n.,* holo: NSW!, iso: NSW!*Halophila ovalis* subsp. *australis* (Doty et B.C. Stone) Hartog, *Verh. Kon Ned. Akad. Wentensch. Afd. Naturk. Sect. 2*, 1970, *59,* 251.

Marine, occasionally estuarine, submerged, perennial herb. Rhizome fleshy, 1–2 mm diam.; scales glabrous. Leaves with petiole to 10 cm long; blades linear-lanceolate to narrowly elliptic 25–70 mm long, 6–15 mm wide; obtuse; cross veins 10–16, branched. Spathal bracts 4–7 mm long. Male flowers with pedicel to 20 mm long at anthesis; tepals 5–7 mm long, imbricate, hooded; anthers oblong, 2.5–3 mm long. Female flowers on erect extended lateral shoot 5–15 cm long, with paired leaf blades 3–5 cm long; lateral shoots sometimes producing successive flowering shoots. Female flowers sessile; hypanthium 2–5 mm long; tepals 0.5–1 mm long; ovary ovoid, 12 mm long; styles 6, 6–20 mm long. Fruits ovoid, 7–13 mm diam. Seeds 40–60, subglobose, 0.5–0.75 mm diam.; testa reticulate.

Endemic to Southern Australia; grows on sand and on mud at depths of 2–35 m. Flowers October to December; fruits November to February.


**Sect. 2. Halophila**
*Halophila* sect. *Barkania* Asch., *Nuovo Giorn. Bot.* Ital., 1871, *3*, 301.Type species: *Halophila madagascariensis* (Steudel) Doty & B.C. Stone.*Barkania* Ehrenb., *Abh. Königl. Akad. wiss. Berlin*, 1832, *1*, 429 (1834). T: not designated.*Halophila* sect. *Typicae* Ostenf., *Bot. Tidsskr*., 1902, *24*, 240. T: not designated.

Marine to estuarine; dioecious. Lateral shoots extremely short, with 2 basal scales. Leaves petiolate, margins entire, glabrous, cross veins present. Flowers on rhizome nodes; style 3.

Around 13 species widely distributed in tropical and subtropical and extending also into warm-temperate waters in the West Pacific Oceans.


**Key to the Species**
**1.** Leaf blade distinctly ovate to elliptic 
**2**
**1:** Leaf blade oblong to spatulate
**6**
**2.** Leaf blade greater than 30 mm in length; cross veins 12–28
**3**
**2:** Leaf blade, less than 25 mm in length; cross veins less than 12
**5**
**3.** Leaf blade with cross veins (16-) 18–28; ½ blade width:distance between intramarginal veins and blade margin ratio 20–25: 1
**H. major**
**3:** Leaf blade with cross veins 12–16; ½ blade width:distance between intramarginal veins and blade margin ratio 12–16: 1
**4**
**4.** Leaf blade with cross veins 12–16 (-18), ascending angle at 40^o^
**H. ovalis**
**4:** Leaf blades with cross veins (8-) 9–14 (-15), sharply ascending at 20–30^o^. Endemic to Fiji, Samoa and Tonga
**H. bullosa**
**5.** Space between leaf margin and intramarginal vein, and space between two adjacent cross veins narrow; cross veins rarely branched
**H. minor**
**5:** Space between leaf margin and intramarginal vein, and between two adjacent cross veins wide; cross veins never branched
**H. gaudichardii**
**6.** Leaf blade with cross veins more than 12
**7**
**6:** Leaf blade with cross veins less than 10
**9**
**7.** Blade extremely linear, 8–16 mm length, width 1.5–3 mm, L:W ratio 6–20:1
**H. linearis**
**7:** Blade elongate to spatulate, more than 30 mm in length, more than 4 mm in width; L:W ratio 7–8:1
**8**
**8.** Blade 35–40 mm long, 4–7 mm wide, L:W ratio 5–7:1; restricted to estuarine
**H. ramuramathiana**
**8:** Blade up to 30 mm long, 7 mm wide, L:W ratio 3–4: 1; endemic to Tanegashima, Japan
**H. mikii**
**9.** Intramarginal space, 0.5–1 mm wide. Endemic to temperate Japan and Southern Korean Peninsular
**H. nipponica**
**9:** Intramarginal space narrow, less than 0.3 mm
**10**
**10.** Blade spatulate to narrowly obovate
**11**
**10:** Blade elliptic to oblong not spatulate
**12**
**11.** Blade 12–16 mm long, 1.5–4.5 mm wide, L:W ratio 3.5–4; cross veins 6–7
**H. okinawensis**
**11:** Blade 20–30 mm long, 2.5–6 mm wide; cross veins 10–16. Endemic to Hawaii Islands
**H. hawaiiana**
**12.** Blade 5–25 mm long, 1–4 mm wide; cross veins 5–8. Intramarginal space 0.15 mm. Endemic to Southern Florida
**H. johnsonii**
**12:** Blade less than 15 mm in length; 3 mm width; cross veins 7, no branched; intramarginal veins extremely narrow, 01–0.2 mm.
**H. madagascariensis**


**1. Halophila major** (Zoll.) Miq., *Fl. Ind. Bat*. 1855, *3*, 230. (Appendix A).T: Indonesia, Sumbawa, Kambing in the Bay of Bima, Oct 1847, *H Zollinger 3430* (lecto: P, iso-lect BO, L, U, Z, *fide*: Kuo et al. 2006: 136–137).*Lemnopsis major* Zoll., *Syst. Verz*., 1854, *1*, 75. T: not designated.*Halophila ovalis* var. *major* Asch., *Linnaea*, 1868, *35*, 174. T: not designated.*Halophila euphlebia* Mak., *Bot. Mag. Tokyo*, 1912, *26*, 208, Figure XV. T: Japan, Shikoku Is., Tokushima Pref., Haifu Gun, Shishikui Mura, 22 Aug 1906, *S Nikai 1613*; holo: TI!, iso: NS!.

Rhizomes robust, fleshy, internodes (20-) 25–40 (-50) mm long, 1.5–2 mm diam. Petioles scale 3.5–4.5 × 1.5–2 mm, rhizome scale 4–5.5 × 1.5–2.5 mm. Petioles (10-) 15–30 (-35) mm long. Leaf blades, ovata, oblong to elliptic, (13-) 15–25 (-30) mm long, (7-) 9–11 (-13) mm wide, L:W ratio 1.4–3:1; apex acute or acuminate; base cuneate to attenuate, symmetrical. Cross veins distinct, (16-) 18–28, often branched (up to 3 branches), distance between adjacent cross veins 0.7–1.25 mm. Distance between intramarginal veins and blade margin 0.16–0.5 mm; 1/2 blade width:distance between intramarginal veins and blade margin ratio 20–25:1. Male flowers, spathal bracts 5–6 mm long; tepals 3–4 mm long; anthers 2.5–3 × 1.2–1.5 mm. Female flowers spathal bracts 5.5–6 mm long; ovary ovoid, white, 1.5–2 × 1.5 mm; styles 25–40 mm long. Fruits ovoid, 4 mm diam., persistent beak up to 8 mm long. Seeds with testa reticulate.

This species usually co-exists with *H. ovalis* and/or *H. minor* and is widely distributed in the Indo-Pacific Oceans to the Red Sea. *H. major* grows in shallow water but also grows down to 6–7 m in the water depth. The flowering and fruiting of this species usually occur during two- to three-month periods between June and December.

**2. Halophila ovalis** (R. Br.) Hook. *f. Fl. Tasman*, 1858, *2*, 45. (Appendix A).T: Australia, Queensland, at the inner entrance to Thirsty Sound (22°20′ S; 149°55′ E) southern part of Long Island, 24 Sept 1802, *R Brown 5816*, holo: K!; iso: BM!*Caulinia ovalis* R. Br. (1810) *Prod. Fl. Nov. Holl.,* 1810, *1*, 339. T: not designated.

Perennial. Rhizomes internodes to (15-) 25–40 mm long, 0.8–1 mm diam. Petiole scale 4 × 1.5–2 mm, rhizome scale 3.5–4 × 1.5–2 mm. Petioles fleshy, purple to red, (5-) 10–25 mm long. Leaf blade bright green, ovate, elliptic, (10-) 12–18 mm long, 4–8 (-10) mm wide, L:W ratio (1.3-) 1.6–2.0:1; apex acute or acuminate; base cuneate to attenuate, symmetrical; distance between intramarginal veins and blade margin 0.25–0.4 mm wide; 1/2 blade width:distance between intramarginal veins and blade margin ratio 10–16:1; cross veins 12–16 (-18), often branched, distance between cross veins 0.8–1.1 mm wide. Male flower spathal bracts 3–4 mm; tepals 2.5–3.5 mm long; pedicel to 20 mm long at anthesis; anthers 2–2.5 mm long. Female flower spathal bracts 3.5–4 mm long; ovary ovoid, white, 1–1.5 × 0.5 mm; hypanthium 4–6 mm long; styles 8–30 mm long. Fruit ovoid, 2.5 × 2 mm. Seeds 15–30, ovoid, ca. 1 mm diam., testa reticulate.

*Halophila ovalis* is widely distributed in the tropical Indo-Western Pacific Oceans but does not occur in the Mariana Is. It grows from low water to 48.5 m deep on various substrates.

**3. Halophila bullosa** (Setch.) J. Kuo, n. comb. (Appendix A).*Halophila ovalis* var. *bullosa* Setch.: Setchell, *Carnegie Inst. Washington Pub*. 1924, *341*, 114–115. Figure 6. T: Samoa, Tutuila Island. Pago Pago Harbour, Aua Beach, ca 1 m, in white coral sand, patches, 12 June 1920, *WA Setchell 114* (lecto: UC!; isolecto: BISH!, C!, S! US! *fide*: Smith, 1979, 128).*Halophila ovalis* ssp. *bullosa* (Setch.) Hartog, *Verh. Kon Ned. Akad. Wentensch. Afd. Naturk. Sect. 2*, 1970, 251.

Submerged marine herbs, rhizomes internodes (10-) 15–30 (-45) mm long, (0.4-) 0.8–1.2 mm diam.; scales convolute, oblong, truncate at base, retuse at apex. Petioles (5-) 10–20 (-30) mm long. Leaf blades elliptic to obovate, apex acute or acuminate, base cuneate, symmetrical, 10–22 mm long, 5–10 mm wide, L:W ratio (1.4-) 2–3 (-4.8):1; cross veins (8-) 9–14 (-15), sharply ascending at 70–80°; space between two adjacent cross veins 0.8–1.2 mm; space between intramarginal veins and blade margins 0.35–0.5 mm. 1/2 blade width:distance between intramarginal vein and blade margin ratio ca. 5–9:1. Spathes male: 4 × 2 mm, female 5 × 2 mm, convolute. Male flowers stamen 3 mm long; anthers 1.5 mm. Female flowers ovary 1.5 × 0.5 mm; hypanthium 5–6 mm long; styles 10 mm long. Fruits ovoid.

**Note:** The type specimen *Setchell 114* had smaller leaf blades (7–10 × 4–4.5 mm with 7–8 cross veins) and without a predominated bullolate appearance. 

*Halophila bullosa* is restricted to Fiji, Samoa and Tonga Islands. The species grows at mud-slit sediment, around 0.5 m in LSW, as isolated small patches in sheltered bays. T Powell’s collections (at K) suggest that flowering and fruiting occur in April and May.


**4. Halophila minor** (Zoll.) den Hartog, *Fl. Males*. I, 1957, *5*, 410, f. 17b. (Appendix A).T: Indonesia, Lesser Sunda Islands, Flores, near Bari, 12 July 1847, *H. Zollinger 3334*, holo: BM; iso: P, L, *fide*: Kuo et al. *Acta Phytotax. Geobot*., 2006, *57*, 134.*Lemnopsis minor* Zoll., in *Zoll. Syst. Verz*., 1854, *1*, 75. T: not designated.*Halophila lemnopsis* Miq., *Fl. Ind. Bat*., 1855, *3*, 230, *nom. illeg*. *Halophila ovalis* var. *minor* (Zoll.) Asch., *Linnaea*, 1868, *35*, 175. T: not designated. 


Rhizome internodes to 30 mm long, 0.5 mm diam. Petiole scale 3 × 1.5–2 mm, convolute, transparent; rhizome scale 3–4 × 1–1.5 mm. Petioles 5−10 mm long. Leaf blades, ovate, to 10 mm long, 4−5 mm wide, L:W ratio 1.8–2.2; apex acute or acuminate; base cuneate to attenuate, symmetrical; distance between intramarginal veins and blade margin 0.15–0.4 mm wide; 1/2 lamina width:distance between intramarginal veins and blade margin ratio 1:11.5–16.3; cross veins 7–12, rarely branched; distance between adjacent cross veins 0.65–0.85 mm. Male flower spathal bracts, 2.5–4 × 1–1.5 mm; tepals 2–3.5 mm long; anthers, oblong, 2.5–3 × 1.2–1.5 mm. Female flower spathal bracts 3.5–4 mm long; ovary elliptic to ovoid to elliptical, 1–1.5 × 0.5 mm; hypanthium 4–6 mm long; styles 8–20 mm long. Fruits 2.5 × 2–3.5 mm. Seeds 15–30, ovoid, 0.5 mm diam., testa reticulate.

The species is widely distributed in the tropical Indo-Western Pacific Ocean. The species usually grows in shallow water with *H. ovalis* and *H. major*. Flowering and fruiting of this species take place April–June in the Philippines; August in Kenya; September to February in tropical Australia.

**5. Halophila gaudichardii** J. Kuo, *Acta Phytotax. Geobot*., 2006, *57*, 145, Figures 8, 15. (Appendix A).T: Marianna Island, *sine loc. exact*, *C. Gaudichard* (lectoholo: P; iso: K-drawings of the type, L 5395235524, *fide*: Kuo *et al*., 2006: 146).*Halophila ovata* Gaudichard in *Freycin., Voy. Bot.*, 1827, Pl. 40, f. 1. T: not designated.*Halophila ovalis* var. *ovata* Asch., *Linnaea*, 1868, *35*, 173. T: not designated.

Rhizome internodes 10–30 mm long, 0.4–0.8 mm diam. Rhizome scale 3 × 0.5–1 mm, petiole scale 3 × 1–1.5mm. Petioles whitish, 10–15 mm long. Leaf blades obovate, (8-) 10–15 (-17) mm long, (3.5-) 4–6 (-8) mm wide, L:W ratio (1.6-) 2–2.4 (-2.6); apex rounded; base rounded; distance between intramarginal veins and blade margin 0.4–0.6 mm wide; 1/2 blade width:distance between intramarginal vein and blade margin ratio ca. 4–8.3:1; cross vein (3-) 4–8, arising at angles of 30°–45°, unbranched, distance between adjacent cross veins 1.75 mm wide. Male flowers with spathal bracts 2.5−3.5 × 1.5 mm; pedicel 4–10 mm long; stamens 3.5–4 mm long; anthers 1.5–3 × 1 − 1.2 mm. Female flowers spathal bracts 4–5 mm long; ovary ellipsoid to ovoid, white, 1.5 mm long, 1 mm wide; style 6–20 mm long. Fruits ovoid, 2–4 × 1.5−3 mm. Seeds 6–30, ovoid, ca. 0.2–0.5 mm diam., testa reticulate.

*Halophila gaudichardii* has been collected from Saipan, Guam and Yap in the Mariana Islands and in the Philippines. However, the species is not found in Caroline Islands, which are adjacent to the Mariana Islands in the Pacific Ocean. This species in the Mariana Islands and the Philippines grows in the shallow water, while it has been collected from 24 m in the Okinawa Is. Flowering and fruiting occur from January to April.

**6. Halophila ramamurthiana** (Ravikumar & Ganesan) J. Kuo, n. com. (Appendix A).*Halophila ovalis* (Br.) Hook. *f*. ssp. *ramamurthiana* Ravikumar & Ganesan, *Aquat. Bot.* 1990, *36*, 351.T: India, Tamil Nadu State, South Arcot, district, Marakkaam (Kaliveli Tank), to a depth of ca. 2 m, 29 March 1987, *Parthasarathy & Ravikumar 85440* (CAL, MH, not seen).

Intertidal to submerged estuarine rarely marine herbs, rhizomes internodes up to 35 mm long, 0.5 mm diam. Petiole scales, 5.5–7 × 3–5 mm, rhizome scales oblong, 7 × 4 mm; petioles up to 60 mm long. Leaf blades oblong, never elliptic, apex acute, cuneate to attenuate and symmetrical at base, up to 35–40 mm long, 4–7 mm wide, L:W ratio 5:1, cross veins 12–16, rare branched, distance between adjacent cross veins 1–1.8 mm wide; intra-marginal veins 0.3–0.45 mm from the blade margin. Male spathal bracts, ovate-lanceolate, 5 × 3 mm, hyaline, tepals, ovate-oblong, 2.5–3 × 2–2.5 mm; stamens 2.75 mm long, anther oblong 2.5–3 × 0.5 mm. Female flowers spathal bracts, lanceolate, 6–8 by 3–4.5 mm; ovary ellipsoid to ovoid, 2.75 × 1 mm; hypanthium 7 mm, styles, up to 23 mm long. Fruit ovoid, 2.5–4 × 1–2 mm. Seeds 6–12 (-18) globose, ca. 0.5 × 0.4 mm.

*Halophila ramamurthiana* occurs in geographic isolated brackish waters of the Coromandel Coast in India and also in Sri Lanka. The plants grow on soft mud in the brackish waters at 1–2 metre depth. They form isolated patches to pure stands of meadows and may mix with *H. ovalis* and *Halodule uninervi*s. This species is not found in the open sea. Flowering mainly occurs from June to September.

**7. Halophila mikii** J. Kuo, *Acta Phytotax. Geobot*, 2006, *57*, 140, Figures 5, 13. (Appendix A).T: Japan, Kagoshima Pref., Tanegashima Island, Kumage Gun, Sumiyoshi, 27 Oct 1921, *Z. Tashiro* (KYO).

Rhizome internode up to 55 mm long, petiole up to 30 mm long. Leaf blade obovate to spatulate, to 30 mm long and 7 mm wide, L:W ratio 3–4:1; apex obtuse; cross veins 14–17, some branched, distance between adjacent veins 1 mm wide; distance between intramarginal veins and blade margin 0.2 mm wide; ½ blade width:distance between intramarginal veins and blade margin ratio ca. 15:1. Female flower with ovary 1.5 mm long; hypanthium 8 mm long. Male flowers and fruits not seen. 

*Halophila mikii* is only known from Tanegashima and Yakushima in Southern Japan. The species grows as small patches at 2 to 10 m on mud substratum. Flowering and fruiting may take place from August to October.

**8. Halophila linearis** den Hartog, *Act. Bot. Neerl*. 1957, *6*, 46, F. 1. (Appendix A).T: Mozambique, Inhaca Is., west coast, exposed at spring ebbtide, July 1937, *E Cohen 20652* (holo: BM! iso: L!)*Halophila ovalis* ssp. *linearis* (Hartog) Hartog: *Verh. Kon Ned. Akad. Wentensch. Afd. Naturk. Sect. 2*, 1970, *59,* 251.

Submerged herbs, rhizomes internodes (10-) 15–40 (-45) mm long, (0.5-) 0.8–1.2 (-1.5) mm diam. Petioles (5-) 10–20 (-30) mm long. Leaf blades linear, oblong, never elliptic; apex acute or acuminate, base cuneate to attenuate and symmetrical (15-) 18–25 (-30) long, (0.8-) 1.5–3 mm wide, L:W ratio (5-) 10–15 (-20); cross veins (10-) 14–16, unbranched, space between adjacent cross veins 0.5–0.8 mm; space between intramarginal veins and blade margin 0.15 mm. Male flowers spathal bract ovate, 3–5 mm long; tepals 2.5–3 mm long, 1–1.2 mm wide, oblong to elliptic. Female flowers spathal bract 4–6 mm long; ovary ovoid, 1–1.5 mm diam.; styles 10–15 mm long. Fruits ovoid to globose, 3–4 mm long, 2–3 mm wide, persistent beak 2–5 mm long. Seeds (4-) 10–12, subglobose; testa reticulate. 

*Halophila linearis* apparently endemic to Mozambique, it occurs in shallow water, sometimes with *H. ovalis* at Inhaca Island and Maputo in Southern Mozambique, as well as at Inhambane in Northern Mozambique. Flowering in August to September and fruiting takes place October to December.

**9. Halophila nipponica** J. Kuo, *Acta Phytotax. Geobot*., 2006, *57*, 141, Figures 6, 13. (Appendix A). T: Japan, Chiba Pref., Okinoshima, Teteyama, 12 Aug.,1994, *Y. Hirano s.n*. (TNS- holo; TI, MAK, UWA-iso).*Halophila japonica* Uchimura et Faye, *Bull. Natn. Sci. Mus. Tokyo*, *Ser. B*., 2006, *32***,** 131, Figs. 1–12. T: Japan, Kanagawa Pref., Yokosuka City, Odawa Bay (35°13′16” N; 139°37′17” E), 7 June 2005, *M. Uchimura s.n*. holo, iso.: TNS (not seen).*Halophila nipponica* subsp. *notoensis* Ohba et Miyata, *Seagrasses of Japan*, 2007, 15, 103, Pl. 68, 69.T: Japan, Ishikawa Pref., Kashima-gun, Notojima-machi, Sowaji, 14 Oct 2005, *T Ohba & M Miyata s.n.* (holo: CBM, not seen). No Latin description.*Halophila* x *tanabensis* Ohba et Miyata, *Seagrasses of Japan*, 2007, 17, 107, Pl 73. No Latin description.T: Japan, Wakayama Pref., Shirahama-cho, Eura, 5 July 2006, *M Miyata* (holo: CBM, not seen).

Rhizomes fleshy, white, internodes 15–35 mm long, 0.8–1.2 mm diam. Petiole scale 3.5−4.5 × 1.5−2 mm, rhizome scale 3.5–4 × 1–1.5 mm. Petioles 5–30 mm long, 0.5–0.8 mm diam. Leaf blade elliptic, (12-) 18–25 (-30) mm long, (3-) 5–9 (-13) mm wide, L: W ratio (2-) 3–4 (-6):1; apex apiculate, obtuse to rounded; base cuneate, symmetrical. Distance between intramarginal veins and blade margin 0.5–1 mm wide; 1/2 blade width:distance between intramarginal veins and blade margin ratio 1:1.5–6.5; cross veins not distinct, (6-) 7–9 (-10), unbranched, distance between adjacent cross veins 1.4–2 mm. Male flower spathal bracts 5 mm long; tepals bright yellow, 3–4 × 1.5 mm; pedicel extended to 6–8 mm long, 0.5–0.8 mm thick; anthers purple, 1.5–2 mm long, 0.2 mm wide. Female flower ovary 1.5–2 × 1.5 mm; hypanthium 3.5–8 mm long; styles 15–25 mm long. Fruit ovoid, 3.5–4 × 3–3.5 mm, persisting beak 3.5–8 mm long. Seeds 25–30, ovoid, ca. 0.6 × 0.7 mm bluntly beaked at both ends; testa reticulate. 

The species occurs in the temperate coasts of Honshu Is. Japan and southern coast of Korean Peninsula. The plants grow on mud-slit, sand sediment, around 5–10 m in LSW as isolated small patches or with *Zostera* in sheltered bays. Flowering in August–September.

Ohba and Miyata (2007) named a subspecies *H. nipponica* ssp. *notoensis* for the population in the Sea of Japan. These authors also described a hybrid as *Halophila* x *tanabensis* from Wakayama, Japan.

**10. Halophila okinawensis** J. Kuo, *Acta Phytotax. Geobot*., 2006, *57*, 144, Figures 7, 14. (Appendix A).T: Japan, Okinawa Pref., Okinawa Isl., ca. 100 m from causeway, Southern Kin Bay, 8–9 m, 22 May 1997, *Z. Kanamoto s.n.*, holo: TNS; iso: TI, MAK, UWA.

Rhizome internodes thread-like thin, to 40 mm long, 0.2–0.3 mm diam., rhizome scale 2.5–3 × 0.5 mm, petiole scale 3–3.5 × 1 mm. Petioles 5–15 mm long. Leaf blade distinctly linear to spatulate or narrowly elliptic; slightly abaxially curved inwards; 12–16 mm long, 1.5–4.5 mm wide, L:W ratio (2.5-) 3.5–4 (-7):1; apex rounded; base asymmetrically narrowed; distance between intramarginal veins and blade margin 0.2–0.3 mm wide; 1/2 blade width:distance between intramarginal vein and blade margin ratio ca. 2–9.5:1; midrib and intramarginal veins distinct, with protrusions; cross veins indistinct, (5-) 6–7, unbranched; distance between adjacent cross veins ca. 2 mm wide. Spathal bracts 3–5.5 mm long, lanceolate, acute. Male flower with pedicel to 20 mm long, 0.45 mm wide at anthesis; tepals 2.5–3 mm long, 1.5–2.5 mm wide; anthers 2.5–3 mm long. Female flower ovary ellipsoid to ovoid, 1.5 mm long, 0.5 mm wide; hypanthium 4–6 mm long; style to 30 mm long. Fruits ovoid, 3 mm diam., persistent beak up to 3.5 mm long. Seeds ca. 15, ovoid, 0.5 mm diam., testa reticulate.

*Halophila okinawensis* has been collected from several localities in Okinawa Island. The species grows in 4 to 24 m deep on sand, muddy sand and coral sand. It grows as mono-specific sparse patches or with *H. decipiens* and *H. gaudichardii* in deeper water and with *Halodule pinifolia* in shallower water. The flowers (both pre and post anthesis) of *H. okinawensis* have been collected in April to May from Okinawa, Japan.

**11. Halophila hawaiiana** Doty *et* B.C. Stone, *Brittonia*, 1966, *18*, 303–305, Figures 1. (Appendix A).T: U.S.A., Hawaiian Is., Maui Is., Kihei, 9 Sept. 1955, *T Matsus. s.n.* (Holo. BISH, lost.Hawaiian Is., Maui Is., Kalama Park, ca. 100 from shore, ca. 1 m of water at low tide, binds sand, 26 Aug 1955, *MS Doty 13009* (L, neotype here designated).*Halophila ovalis* ssp. *hawaiiana* (Doty & Stone) Hartog, Hartog, *Verh. Kon Ned. Akad. Wentensch. Afd. Naturk. Sect. 2*, 1970, *59,* 251. 

Submerged, rhizomes internodes (10-) 15–35 (-50) mm long, (0.3-) 0.8–1.2 (-1.5) mm in diam.; scales at apex, 5 × 3 mm, convolute, at apex, 5 × 2 mm. Petioles 10–25 (-35) mm long. Leaf blades spatulate to very narrowly obovate, never elliptic or ovate; apex acute or acuminate, base cuneate to attenuate, symmetrical merging into petiole, (10-) 15–20 mm long, (1-) 2–4 (-5) mm wide, L:W ratio 3.3–10:1; cross veins 10–16, unbranched; distance between two adjacent cross veins 0.8–2.5 mm; distance from intramarginal veins to blade margin 0.15–0.2 mm. Spathal bracts broadly lanceolate, 5–6 mm long, 2–3.3 mm wide. Male flowers, pedicel 10–25 mm long at anthesis; tepals broadly elliptic, 3.5 mm long, 2.5 mm wide; anthers oblong 1.5–3 mm long, 0.8–1.0 mm wide. Female flowers with ovary 1.2–1.6 mm long, 1 mm wide; hypanthium 4–5 mm long; styles 12–15 (-18) mm long. Fruit ovoid, 4 × 3.5 mm. Seeds 15–30, ovoid, ca. 0.6 × 0.7 mm, testa reticulate. 

*Halophila hawaiiana* is restricted to certain islands in the Hawaii Islands (Oahu, Maui, Molokai and Midway) and grows at low water to 4 m on sands in bays or fishponds. This species exhibited morphological variations from different localities/islands, which McDermid et al. (2003) attributed to the influence of environmental conditions and showed it to have the same DNA (ITS) sequence analyses.

**12. Halophila johnsonii** Eiseman, *Aquat. Bot*., 1980, *9*, 15–19, Figure 1. (Appendix A).T: U.S.A., Florida, Bessie Cove, Hutchinson Island, Martin Co. (27°15’ N;80°12’ W), 23 May 1975, *NJ Eiseman 3411*, holo: HBFH; iso: HBFH, TEX, US (not seen).

Rhizomes slender, internodes 10–20 (-35) mm long, 0.5–1 mm diam. Petiole scale 5 × 3 mm, rhizome scale 5 × 2 mm. Petioles (10-) 20–45 mm long. Leaf blade linear, apex acute or acuminate, slightly asymmetrically; base cuneate to attenuate, asymmetrically, tapering gradually into the petiole, (10-) 13–17 (-25) mm long, (1-) 2–3 (-4) mm wide, L:W ratio (4.5-) 6–8 (-10):1; midrib very prominent; cross veins (5-) 6–7 (-8), unbranched, space between adjacent cross veins *ca* 1.25 mm. Space between intramarginal veins and blade margin 0.15 mm; ½ blade width:distance between intramarginal veins and blade margin ratio ca. 10–20:1. Spathal bracts lanceolate, keeled, apex acuminate, 5 × 2 mm, convolute. Flowers and fruits not seen. 

*Halophila johnsonii* is apparently restricted to the costal lagoons of Indian River, Eastern Florida, from Virginia Key (25°45’ N) in Biscaynbe Bay to Sebastian Inlet (27°51’ N). *Halophila johnsonii* is the only member of the section Halophila occurring in the American continent and can be considered as a relic. The species has been considered as asexual species, because neither male flowers nor fruits have been found.

**13. Halophila madagascariensis** Steud. ex Doty *et* B.C. Stone, *Taxon*, 1967, *17*, 417, Figure 1. (Appendix A).T: Madagascar, exact locality unknown (in littoris orientalis sbmarinus), *Du Petit Thouars* (holo: P!, iso: BM!).*Halophila madagascariensis* Steud., *Nom*. ed. 2, 1840, *1*, 720 (*nomen*).

Rhizomes internodes 0.5–1 mm diam., up to (5-) 10–35 (-45) mm long. Petioles (5-) 10–30 (-45) mm long. Leaf blades oblong to narrowly elliptic, apex acute or acuminate, base cuneate to attenuate, symmetrical, 8–13 (-16) mm long, (3-) 4–6 (-8) mm wide, L:W ratio (2-) 2.5–4 (-5):1; cross veins (6-) 7–12, ascent at 40° angle, occasionally branched; distance between adjacent cross veins 1–1.2 mm. Distance between intramarginal veins and blade margin 0.1–0.2 mm. Spathes lanceolate, keeled, apex acuminate, 5 × 2 mm, convolute. Flowers and fruits are not seen.

*Halophila madagascariensis* is widely distributed in islands of the West Indian Ocean, does not extended into the east coast of African continent, nor to the Indian Peninsular. The plants grow around 0.5 to 1 m in LSW, as isolated small patches in sheltered bays. Little is known about reproductive biology; only one specimen (out of 30 collections) had reproductive material with young male flowers.


**Sect. 3. Stipulaceae**
*Halophila* sect. *Stipulaceae* J. KuoType species: *Halophila stipulacea* (Forrsk.) Asch.

Dioecious. Petioles sheathing lopsidedly; leaf blade margin serrulate, surface glabrous, slightly hairy or papillose. 

Dioica. Petioli impariter vaginati; foliorum lamina serrulata, glabra vel parum hirta vel papillosa.

The most distinctive characteristic in this section is a pair of large persistent, transparent scales covering the short petioles. Two species in the section, *H. stipulacea* originally from the Red Sea, immigrated to the Mediterranean Sea after the opening of the Suez Canal, recently also found in the Caribbean Sea. *H. balfourii* is confined to Mauritius Is. in the Indian Ocean.


**Key to the species**
**1.** Leaf blade up to 60 cm long, 10 mm wide, surface glabrous, minute hairs, occasionally bullate, cross veins 10–40 
**1. H. stipulacea**
**1:** Leaf blade 35 mm long, 5 mm wide, surface papillous, not hairy, bullate; cross veins less than 14
**2. H. balfourii**


**1. Halophila stipulacea** (Forrsk.) Asch. *Sitz.-Ber. Ges. Natirf. Fr. Berlin,* 1867, 3. (Appendix A).T: Neotype: Eritrea, Massaua, *G. Schweinfurth 8*, 26 Jan 1891(G), isoneotype (P), *file*: Ferrer-Gallego & Boisset, *Taxon*, 2015, *64*, 1935.*Zostera bullata* Délile, *Fl. Aegypt*. **1**813, *Ill*, 75, 145, Pl. 53, f.6.T: Lect.: ‘a Suez sur la sable, la plante croissant au fond de la me rest verte et a la port d’un potamogrton (MPU), *file:* Ferrer-Gallego & Boisset, *Taxon*, 2015, *64*, 1935.

Marine, rather robust species with large elliptic or obovate transparent scales, 12–17 mm long, 6–10 mm wide, folded at the rhizome nodes covering the short petioles. Blades linear to oblong, elliptic, membranes; surfaces glabrous or with minute hairs, occasionally bullate, not papillous; up to 60 mm long, 10 mm wide; cross veins 10–40, branched. Male flower with tepals 3–4 mm long; anthers 2–3.5 mm long. Female flowers with 3 styles 20–25 mm long. Fruits ellipsoid 3–4 mm long. Seeds testa reticulate in squares appearance. 

*Halophila stipulacea* is the most successful migrating species among the *Halophila* species. The species migrated from the Red Sea to the Mediterranean Sea after the opening of Suez Canal and recently spread to the Caribbean Sea. Flowering and fruiting in May to October.

**2. Halophila balfourii** Solereder *Beih. Bot. Centralbl*. 1913, *1*, 47. (Appendix A).T: Lectotype: Rodriguez, Dr. I.B. Balfour. Aug.-Dec. 1874 (BM, isolecto. P, L, K), *file*: Ferrer-Gallego & Boisset, *Taxon*, 2015, *64*, 1035.

Submerged herbs, rhizomes (0.2-) 0.5–0.8 (1.2) mm diam., internodes (5-) 10–30 mm long, with one root at least 60 cm long. Scales two, elliptic or obovate, white or transparent, glabrous, margins entire; the petiole scale, apex retuse, base ovate to oblong, subtruncate, 8–10 mm long, 2.5–3 mm wide; the rhizome scale, apex retuse, base oblong, truncate, convolute, hyaline 10–12 mm long, 2–3 mm wide. Petioles (3-) 5–10 mm long. Leaf blades distinct elliptic, cartilaginous, papillose, not membranous, nor bullate, (15-) 20–30 (-35) mm long, (1.5-) 2–4 (-5) mm wide, L:W ratio 7–12:1; apex distinctly pointed, base cuneate, symmetrically into the petiole; margin serrulate, especially on the apical region; cross veins (6-) 10–12 (-14), unbranched, ascending at angles around 45^o^. Space between blade margin and intramarginal veins, extremely narrow ca. 0.1 mm. Male flowers scales 4 mm long, 2.5 mm wide; pedicel thin, up to 10 mm long, 1 mm diam. at anthesis; tepals 3–4.5 mm long, 2–2.5 mm wide. Female flowers with ovary ovoid to ellipsoid, 1.5–2 mm long, 1–1.5 mm wide; styles 3, 15–25 mm long. Fruits ellipsoid (2-) 3.5–4 mm long, (2-) 3 mm wide. Seeds (12-) 20–30, globular, 0.75 × 0.5 mm, at both ends contracted; testa reticulate with a rectangular appearance. 

*Halophila balfourii* is confined to Rodriguez and Mauritius Islands. The species has been collected from less than 1 m to 6 m in the depth. Both male and female flowers occur in October and fruits are produced from October to December in Mauritius.


**Sect. 4. Decipientes**
*Halophila* sect. *Decipientes* J. KuoType species: *Halophila decipiens* Ostenf.

Monoecious. Male and female flowers form on the same or different floral nodes. Leaf blade margin serrulate, surface hairs, spines or glabrous. 

Monoica. Flores masculi et feminei ad genicula floralia eadem vel discreta facti. Foliorum lamina serrulata, hirta, spinosa vel glabra. 

Three species in the section, *H*. *decipiens* occurs worldwide. The other two species are restricted to the West Pacific region.


**Key to the species**
**1.** Male and female flowers on the same spathes; petiole shorter than blade
**1. H. decipiens**
**1:** Male and female flowers on separate spathes of the same rhizome (plant)
**2**
**2.** Petiole shorter than the blade; blade surface usually with stiff hairs
**2. H. capricorni**
**2:** Petiole as long as the blade; blade surface glabrous
**3. H. sulawesii**


**1. Halophila decipiens** Ostenf. *Bot. Tidsskr.* 1902, *24*, 260. (Appendix A).T: Off Kohdat, Gulf of Thailand, Feb, 1900, *J. Schmidt 540*; holo: C!; iso.: L!.*Halophila decipiens* var. *pubescen*s Hartog, Fl. Males., 1957, *5*, 411. T: not designated.

Marine to estuarine, submerged, annual or perennial herb. Rhizome fleshy, c. 1 mm diam.; scales usually hairy. Leaves with petiole distinctly shorter than blades. Blades oblong-elliptic, 10–25 mm long, 2.5–6.5 mm wide, finely serrulate, obtuse to acute, membranous, hairy or sometimes glabrous; cross veins 5–9, unbranched. Spathal bract hairy, enclosing a male and a female flower. Male flowers with pedicel to 25 mm long; tepals ovate to elliptic, 2 mm long; anthers linear-oblong, 1.5–2 mm long. Female flowers with hypanthium 1–2 mm; tepals minute; ovary ovoid, 1 mm long; styles 3, 1.5–2.5 mm long. Fruits ellipsoid, 2–5 mm diam. Seeds c. 30, ovoid, 0.2–0.5 mm diam.; testa reticulate.

*Halophila decipiens* is the most widely distributed among the *Halophila* species, occurring in tropic to temperate regions of all major oceans. It grows from shallow water to 58 m and also associates with mangroves and estuaries. Flowers in summer, fruits in summer to autumn.

**2. Halophila capricorni** Larkum, *Aquat. Bot*., 1995, *51*, 320. (Appendix A).T: Australia, Steve’s Bommie, One Tree Is., Great Barrier Reef, Qld., 13 Nov. 1990, *A.W.D. Larkum s.n.*; holo: NSW; iso: AD, SYD, UWA.

Marine submerged, perennial herb. Rhizome fleshy, 0.9–1.5 mm diam.; scale obovate, 5–8 mm long, hairy, keeled. Leaves with blade longer than petiole; blade obovate to oblong-elliptic, 15–30 mm long, 5–9 mm wide, finely serrulate, rounded at apex, not membranous, with few stiff hairs; cross veins 9–14, occasionally branched. Spathal bracts hairy, 5–10 mm long, enclosing either male or female flower. Male flowers with pedicel 5.5–7.5 mm long; tepals ovate, 3.5–4.1 mm long; anthers linear, 2.5–3.1 mm long. Female flowers sessile on rhizome nodes; hypanthium 2–3 mm long; tepals 0.2 mm long; ovary 1 mm long; styles 3, 15–25 mm long. Fruit globose, 2.5–5 mm diam. Seeds 18–30, ovoid, 0.5–0.6 mm diam.; testa reticulate. 

*Halophila capricorni* occurs on the Great Barrier Reef and also in New Caledonia, a herbarium (K) record from Indonesia. Grows on coral sand, on the lee side of the reef at depths of 7 to 42 m. Flowers from October; fruits from January to March.

**3. Halophila sulawesii** J. Kuo, *Aquat. Bot*. 2007, *87*, 171. (Appendix A).T: Indonesia, southern Sulawesi, the Supermodel Archipelago, Samalona Is., north side, 15 m, 4 Aug. 1989, *E. Verheij 0388* (L, holo, iso).

Marine. Submerged. Rhizome slender, 0.3–0.8 diam., internode up to 50 mm long. Petioles (5-) 10–25 mm long. Leaf blades ovate to slightly elliptic; margins finely serrulate; surface glabrous; cross veins 12–14 (-16), rare branch. Male flowers with pedicel 1.5–2 mm long; pedicels up to 25 mm long at anthesis. Female flowers with peduncle 2–4 mm long; spathes 7–8.5 mm long, 1 mm wide; ovary ellipsoid, 1.5–2 mm long; hypanthium 7–10 mm long; styles 3, 10–30 mm long, unequal. Young fruits 4 mm long 2.5 mm wide; mature fruits not seen. 

Currently, this species is only known from Spermondel Archipelago, Sulawesi Island, Indonesia and grows 10–30 m on coral sands. Flowering and fruiting occur from August to December.


**Sect. 5. Microhalophila**
*Halophila* sect. *Microhalophila* Asch., *Nuov. Giorn. Bot. Ital*., 1871, *3*, 302.Type species: *Halophila beccarii* Asch.*Halophila* sect. *Pusillae* Ostenf., *Bot. Tidsskr*., 1902, *24*, 240.Type species: *Halophila beccarii* Asch.

Delicate, minute, monoecious plant with 2 scales at the base of erected lateral shoot, which bear a pseudo-whorl of 4–10 leaves at the top. Blades without cross veins, margins slightly serrulate or entire, surface glabrous. Male and female flowers form on different floral shoots of the same plant.

**Halophila beccarii** Asch., *Nuov. Giorn. Bot. Ital*., 1871, *3*, 302. (Appendix A).T: Malaysia, Borneo, Sarawak, in the mouth of the river Bintula, *O. Beccari 11826* (= *P.B. 3666*).

Thin rhizomes with 2 scales covering the base of the erect stem bearing a group of 4–10 leaves at the top. Blades lanceolate, up to 13 mm long, 1–2 mm wide, with no cross veins; apex pointed. Male flowers up to 1 cm long; spathes ca. 4 by 2 mm, pedicel up to 8 mm long; tepals ca. 2.5 by 1.5 mm; anthers linear-oblong, 2 by 0.25 mm. Female flowers up to 2 cm long; hypanthium 2 mm long; styles 2–3, 10–15 mm long. Fruits ellipsoid to ovoid, 4–5 mm by 2 mm. Seed 1–4, globose, 1.2 by 1 mm, testa reticulate. 

*Halophila beccarii* is widely distributed in the Bay of Bengal and the South China Sea. Taiwan is the northern limit of this species. The species is usually associated with mangrove communities and often exposed at low tide. The lack of cross veins is a unique characteristic among *Halophila* species. Flowering and fruiting occur from January to March and also from August to October.


**Sect. 6 Americanae**
*Halophila* sect. *Americanae* Ostenf., *Bot. Tidsskr*., 1902, *24*, 260.Type species: *Halophila engelmannii* Asch.

Perennial, dioecious plants with 2 scales at the base and 2 scales around halfway up the erect lateral shoots, bearing a pseudo-whorl 4–8 leaves at the top. Blades ovate to oblong, cross veins present, glabrous, margins serrulate.

Two species are widely distributed in the Caribbean, the Gulf of Mexico and along the Pacific coast of Central America.


**Key to the species**
1. Petiole extreme short, Blade oblong with pointed tip, cross veins 6–8
**1. H. engelmannii**
1: Petiole 2–5 mm long, blade ovate with a rounded tip, cross veins 3–5
**2. H. baillonis**


**1. Halophila engelmannii** Asch. in Neumayer, *Anl. wiss. Beob*. Reisen, 1875, *1*, 368. (Appendix A).T: U.S.A. Florida, near Apalachicola, deep water, Sept 1865, *Dr. Chapman*, received in a letter from Dr. Gray (NY, lectotypes, here designed). 

Rhizome thin, fragile, with one root at each node, internodes 2–4 cm long. Scales at the base of the lateral shoots broadly obovate, 3–6 mm long; apex obtuse. Lateral shoots 2–4 cm long, bearing around halfway to the top 2 lanceolate to obovate scales, and at the top 2–4 pairs of leaves placed in a pseudo-whorl. Petioles to 2 mm long. Leaf blades oblong or linear-oblong, 10–30 mm long, 3–6 mm in width; apex obtuse, sometimes apiculate; margin finely serrate; cross veins 6–8, ascending at angles of 30–45°. Spathal leaves lanceolate, acuminate sessile in the axiles of a leaf, containing only a flower. Mae flowers with tepals 4 mm long; anthers 4 mm long. Female flowers with a minute perianth; ovary ovoid, 3–4 mm long; hypanthium 3–5 mm long; styles 3, 30 mm long. Fruits globose to subglobose, 3–4 mm diam. Seeds subspherical, 1 mm in diam. testa reticulate.

The species is widely distributed along the coasts of the Gulf of Mexico in Texas and Florida, also in Caribbean Sea.

**2. Halophila baillonis** Asch. ex Dickie in Hook. *f*., *J. Linn. Soc.* 1874, *14*, 317. (Appendix A).T: Virgin Is., St. Thomas, Challenger Expedition, 1873, *Dr. Moseley s.n.* (K).*Halophila aschersonii* Ostenf., *Bot. Tidsskr***.** 1901, *24*, 239. T: St. Croix, Christiansted, Lagune, 26 Feb 1892, *H. Lassen s.n.* (C); St. Croix, Christiansted, Lagune, 1906, *F. Børgese s.n.* (C).

Rhizome thin, fragile, internodes 1–3.5 cm long. Lateral shoots 6–40 mm long, bearing around halfway 2 obovate scales, and at the top 2–3 pairs of leaves placed in a pseudo-whorl. The scales are 3–6 mm long. Petioles 2–5 mm long. Leaf blades oblong, ovate or elliptic to lanceolate, 5–22 mm long, 2–8 mm wide; apex obtuse; margin finely spinulose; cross veins 3–8 ascending at angles of 60–80°. Spathal leaves 5–8 mm long. Male flowers with a pedicel 3 mm long; tepals 4 by 2 mm; anthers oblong, 4 mm long. Female flower 6–7 mm long, with a minute perianth and a sessile ovary gradually continuing into the hypanthium; styles 2 to 5, 10–30 mm long. Fruits globular, 2–3 mm in diam. Seeds 10–20, sub-spherical; testa reticulate.

This species is widely distributed in the Caribbean Sea extending south to Brazil. The plants grow from low water down to 30 m on mud and sands.


**Sect. 7. Spinulosae**
*Halophila* sect. *Spinulosae* Ostenf. *Bot. Tidsskr*. 1902, *24*, 240.Type species: *Halophila spinulosa* (R.Br.) Asch.*Halophila* sect. *Aschersonia* Hartog, *Fl. Males*. I, 1957, *5*, 408.Type species: *Halophila spinulosa* (R.Br.) Asch.

Dioecious. Lateral shoots erect, wiry, with 2 basal nodal scales and numerous nodes. Leaves up to 20 or more pairs on nodes per shoot, sessile, distichously arranged near top of shoot; blade with perpendicular cross veins. Flowers form on apical nodes of erect lateral shoots.

Only one species occurs in tropical and subtropical Malesia and Northern Australia.

**Halophila spinulosa** (R.Br.) Asch. *in* G.B. von Neumayer, *Anl. Wis. Beobacht. Reisen*, 1875, 368. (Appendix A).*Caulinia spinulosa* R.Br., *Prodr. Fl. Nov. Holl.*, 1810, 339.T: Australia, Queensland, *R. Brown Iter Austral. 1815*; holo: BM n.v. (Lost).Neotype: Australia, Queensland, Port Denison, 1871, *F. Kliner s.n.* (MEL 3807); iso-neotypes: Port Denison, 1871, *F. Kliner s.n.* (MEL 3808-3812), here designated.

Marine, submerged, perennial herb. Rhizome 1–1.5 mm dia., scales elliptic or ovate, 3–6 mm long, acute or obtuse. Lateral shoots 20–120 cm long, occasionally branched, with leaf scars from shed leaves. Leaf blade oblong to linear, 5–20 mm long, 1.5–3.5 mm wide, serrulate, glabrous, apex rounded, with basal portion of one side of blade folded upwards; cross veins 4–5. Male flowers shortly pedicellate; tepals elliptic, 3–4 mm long, obtuse, reflexed; anthers linear-oblong, 2.5–3 mm long. Female flowers with hypanthium 5–6 mm; tepals 0.1 mm long; ovary ovoid, 1–2 mm diam.; styles 3–5, 10–12 mm long. Fruits ovoid, 4–6 mm diam. Seeds 20–30, globose, 0.75 mm diam.; testa reticulate.

*Halophila spinulosa is* widely distributed in Malesia and Northern Australia. The species has been found from the level of mean low water spring down to around 45 m on coral platforms, coral sand and muddy sand substrates. Flowers March, October to December, fruits January to March.


**Sect. 8. Tricostatae**
*Halophila* sect. *Tricostatae* M. Greenway, *Aquat. Bot*., 1979, *7*, 67.Type species: *H. tricostata* M. Greenway

Annual, rather delicate herbs. Dioecious. Lateral shoot erect, fleshy, with 2 basal nodal scales. Leaves in pairs at basal nodes, in pseudo-whorls of 3 at distal nodes, sessile; blades with cross veins 2–3. Flowers forming on apical nodes of lateral shoots. One species occurs in NE Australia and the Philippines.

**Halophila tricostata** M. Greenway, *Aquat. Bot*., 1979, *7*, 68. (Appendix A).T: Australia, Queensland, Lizard Is., Cook District, 20–30 m, 1 Dec 1978, *M. Greenway s.n*. (BRI!).

Marine, occasionally estuarine, submerged, annual herb. Rhizome 1 mm diam.; scales suborbicular to obovate, keeled, 3–4 mm long. Lateral shoot up to 80–180 mm long, occasionally branched. Leaf lade linear-oblong, 12–20 mm long, 2–4 mm wide, sparsely serrulate, obtuse, glabrous; midrib conspicuous; cross veins 2 or 3. Male flowers with pedicel to 0.5 mm long; tepals 3–4 mm long; anthers linear-oblong, 2.5–3 mm long. Female flowers with hypanthium 3 mm long, tepals 0.1 mm long; ovary ovoid, 0.5–1 mm diam.; styles 5–6, 10 mm long. Fruit ovoid, 4 mm diam. Seeds 25–60, globose, 0.35–0.45 mm diam.; testa fine papillate.

The plants grow in NE Australia, from Prince Charlotte Bay to Gladstone, also in E Gulf of Carpentaria, between 1.4 and 54 m deep on muddy substrate, near mangrove and on coral sand. The plants also reported to grow in the Philippines. Flowers from September to October, fruits November to January in Australia.

## 3. Discussion

### 3.1. New Sections in the Genus Halophila

The present study retains all four sections used by den Hartog [6] and one section from Greenway [9]. Based on both reproductive and vegetative characteristics, three new sections, section Decipientes, section Stipulaceae and section Australes, are created. All three sections were originally under the section Halophila of den Hartog [6]; thus, the genus *Halophila* contains a total of eight sections in this revision. The section Australes contains a single species, *H. australis*, and its female flowers, with six styles, which are uniquely formed on the extended floral shoots but not on the rhizome nodes. The section Stipulaceae has two species: *H. stipulacea* and *H. balfourii*. They are dioecious and possess unusually large, persisted scales and serrulated leaf blade margins. The section Decipientes contains three species, *H. decipiens*, *H. capricorni* and *H. sulawesii*; all have monoecious flowers with three styles, and leaf blade margins are serrulated. The section Halophila contains the remaining *H. ovalis* with its closely related species. They are dioecious plants with three styles and have long leaf petioles with entire blade margins. The section Microhalophila is monoecious, but on the other hand, the sections Spinulosae, Americanae and Tricostatae all are dioecious. Table 1 shows the species compositions in these eight sections. 

### 3.2. Typifications in Halophila

Den Hartog and Kuo [1] listed *H. madagascariensis* (Steudel) Doty & B.C. Stone as the type species of the genus *Halophila* and declared that it should not be treated as a synonym of *H ovalis*. A complete re-description of this type species is therefore provided in this review.

The earliest collection of *Halophila engelmannii* from Florida is located at NY. In the herbarium sheet, the following was stated: “Deep water, coast of Florida, near Apalachicola, Dr. Chapman, received in a letter from Dr. Gray Sept 1865”. This collection was made prior to Ascherson’s species description in 1875. There are two fragments of plants in the herbarium sheet, each with four internodes with perfect vegetative material. It is determined here that this collection should be designated as lectotypes. Among sixteen paratypes of *H. hawaiiana* cited by Doty and Stone [31], only one specimen (*Doty 13009*, L) was collected from Maui Island, the original type locality. Therefore, this specimen has been selected as neotypes of *H. hawaiiana*. An exhaustive search for the original material of Brown’s *H. spinulosa* has failed to locate any extant specimen. It should be noted that Bentham [59] had mentioned that “the specimen of *C. spinulosa* in Brown’s herbarium have no fructification, but Kliner’s specimens in F. Mueller’s Herbarium in fruit is in other respects precisely similar”. In fact, Mueller [60] had already provided a detailed morphological description of vegetative, female flowering and fruiting information using Kliner’s material. Therefore, it is logical to consider Kliner’s collection as neotypes. Currently, there are four mounted sheets of Kliner’s collection at MEL. The specimen MEL 3807 is a complete specimen that matched the descriptions in the prologue and was also included in the subsequent descriptions of the species by Meuller [60]. Therefore, the specimen MEL 3807 is selected as neotype and MEL 3008, 3009 and 3012 as iso-neotypes of *H. spinulosa*.

### 3.3. New Combinations in Halophila 

Taxonomically, *H. stipulacea* contains two synonyms: *H. bullata* Délile and *H. balfourii* Solereder [6]. The microscopic study carried out in this laboratory reveals that both *H. stipulacea* and *H. bullata* have normal flat blade epidermis, while *H. balfourii* has a unusual papillae epidermis, as described by Balfour [61] previously, which is a unique characteristic among *Halophila* species. Furthermore, *H. balfourii* is restricted to the Mauritius Islands, while *H. stipulacea* occurs in the Red Sea. Therefore, *H. balfourii* has been treated as an independent species, not as a synonym of *H. stipulacea* [62].

Currently, there are three subspecies in *H. ovalis*, vis.: *H. ovalis* ssp. *bullosa, H. ovalis* ssp. *linearis* and *H. ovalis* ssp. *ramamurthiana*. Based on a thorough morphological examination, the present review recommends that they should be treated as independent taxa, although it is open to more vigorous molecular investigation to confirm the status of these little-known taxa. The microscopic studies have shown that the leaf blade of *H. bullosa* has distinct ascending cross veins at 70–80°, which is rather unique among *Halophila* species. Furthermore, most of the leaf blades possess numerous small bullations. Through an isozyme investigation, McMillan and Bridge [63] recommended that this species should be treated as an independent species. However, recent research preferred to treat this species as a subspecies of *H. ovalis* [64]. For *H. ramamurthiana*, unlike a typical *H. ovalis*, it has distinctive oblong membranous leaf blades and the plants were only found in estuarine environments. Nguyen et al. [49] distinguished this species from *H. ovalis* and *H. ovata* in Tamil Nadu, India by using DNA AFLP fingerprint gene marker, but they could not separate these three species using ITS, *rbc*L, *mat*K gene markers. Therefore, based on the strong morphological differences and partial molecular support, as well as its unique habitat, it can be confidently concluded that *H. ramamurthiana* is an independent species. On the other hand, as no molecular or ecological study on the species *H. linearis* has been reported since they were named in 1959, it has been treated as *H*. *ovalis* [65,66]. Since this species has rather unique elongated slender leaf blades and only occurs in the restricted type location of Mozambique, it is suggested to treat this species as an independent taxon until future molecular study evidence becomes available to prove otherwise. On the other hand, as there is no sufficient biological, ecological or molecular information on the *Halophila* type species *H. madagascariensis*, it is recommended that further *Halophila* study should focus on the Southern Indian Ocean. In addition, it can be anticipated that more potential new interesting taxa may be found there. 

### 3.4. Deep Water Halophila

Up to now, the deep-water *H. tricostata*, *H. capricorni*, *H. okinawensis* and *H. sulawesii* have been regarded as endemic to their original type localities in the West Pacific Ocean. In fact, *H. tricostata* had been reported from the Philippines [67] and a juvenile plant of this species had also been illustrated as *Halophila* sp. [68] [p. 43]. A herbarium (K) specimen of *H. capricorni* was collected from Aru Island, Indonesia during Challenger Expedition Moseley, but it was mis-identified as *H. decipiens*. Furthermore, Fortes [68] [p. 41] also illustrated *H. okinawensis* but named it as *H. minor*, a new variety from the Philippines. *H. sulawesii* could easily be considered as deep-water *H. ovalis* [69] because *H. sulawesii* and *H. ovalis* both have ovate leaf blades, except that the former species was collected from 20 m deep water. Furthermore, the former species is monoecious, while the latter is dioecious. The above information suggests that these deep-water *Halophila* species are possibly more widely distributed. Therefore, it is recommended that future deep-water seagrass surveys should be conducted in the West Pacific Region in addition to those carried out in NE Australia [70,71]. 

### 3.5. Species Identification in Halophila

Species identification in most *Halophila* sections is more or less straightforward. On the other hand, some species in section Halophila may have to rely on the combination of several vegetative characteristics such as leaf blade appearance, shape, size and cross vein number, as well as space between blade margin and intra-marginal veins and the space relating to blade width, etc., to present a clear picture of different species’ unique characteristics. Therefore, it is highly recommended to use light microscopes to measure these small structures in detail carefully. 

Currently, section Halophila contains around thirteen described species, indicating that there is a high possibility of morphological plasticity. Some of these species, such as *H. hawaiiana*, *H. johnsonii* and *H. bullosa*, have been considered as conspecific species with *H. ovalis* because there is no genetic diversification among them [43,72,73]. Regardless of the fact that these species are morphologically different and have restricted regional distributions, it has been interpreted that they are different ecotypes of a single species with recent speciation, incomplete lineage sorting or ongoing gene flow through hybridization [74]. Indeed, *H. x tanabensis* as a hybrid between *H. nipponica* and *H. major* was described [56]. More recently, a hybrid between *H. ovalis* and *H. major* has been demonstrated molecularly from Sri Lanka [75]. However, using new sequencing methods recommended by Yu et al. [76] may resolve these conspecific species’ complexity and assist in future seagrass taxonomical research. 

It is true that some species have a vast geographic distribution and occur at different habitats, so when studied more closely, they may turn out to be separate species. For example, *H. major* and *H. ovalis* often grow adjacent to each together and their gross morphology is also very similar. As such, *H. major* has been treated as *H. ovalis* for 150 years. Since *H. major* (synonym as *H*. *euphlebia*) was identified molecularly [77] and morphologically [40] in Japan as an independent taxon, this species has been recorded from several countries in Southeast Asia and the Red Sea [47,50,51,78,79].

The species name used by previous molecular studies could be misrepresented. For example, *H. minor* used by Waycott et al.’s study [43] could in fact belong to *H. gaudichardii*, and *H. ovata* from India in Nguyen’s molecular study [49] could belong to *H. minor* or even represent a new taxon because the species name of *H. ovata* has been replaced by *H. gaudichardii* and this species only exists in the tropical W Pacific region, not India [40]. *H. minor* normally coexists with *H. ovalis* and *H. major* in the shallow waters in W Indo-Pacific regions but does not occur in the Mariana region. Furthermore, a report on the *H. ovalis* from the West Atlantic Ocean [80] should belong to *H. johnsonii*, morphologically. Contrary to Shimada’s molecular conclusion, Kim et al. [46] recognised that both tropical *H. okinawensis* and *H. gaudichardii* are independent species and evolved later than temperate *H. nipponica*.

Currently, the identification of the closely related species in the section Halophila is mainly restricted to the use of vegetative characteristics such as leaf blade appearance and shape and, more importantly, their relationship to cross veins. Due to the high plasticity in this section, some structure characteristics used in key and species descriptions may overlap among closely related species. However, it is strongly recommended to document the reproductive biology of the studied species from local communities, e.g., Muta Harah for the annual *H. beccarii* [80], Short et al. for *H. baillonis* [81], Singh et al. for *H. bullosa* [64]. In addition, it is suggested that the seed coat testa could serve as additional *Halophila* species identification criteria [82]; see also Figure 1. These additional data would improve and strengthen the *Halophila* taxonomy further.

### 3.6. Concluding Remarks

It is anticipated that some of the described *Halophila* species in this review may become synonyms after more studies are conducted in the future. On the other hand, there are potentially more new species in the *Halophila* yet to be described [75,83]. It has been suggested that the new species should be accepted if it is supported by genetic data [84]. However, it is essential to carefully and thoroughly check through the samples used in the molecular studies to make sure that they correctly represent the species to be compared. It is interesting to note that Nguyen et al. [51] reported that the genetics of *H. ovalis* in the Red Sea are different from those of other parts of the world and the Red Sea *H. ovalis* is potentially a new taxon despite the two populations having identical plant morphology. Therefore, it is also very important that these new taxa should have the reproductive information to strengthen the claim.

The present review on the *Halophila* taxonomy is by no means perfect. We should acknowledge that since the plants are so small and useful morphological characteristics for the species identification are so limited, it is highly recommended that all future molecular studies should conduct analyses using the newly developed sequencing methods or invent better markers to analyse multiple samples across the species range, particularly to include specimens from the species type locality. In addition, these studies should check whether the morphological data of the studied taxa are accurate. Therefore, it is encouraged to apply microscopic technologies to provide detailed information on both vegetative and generative structures to ensure that the correct identity of each species studied is verified. It is particularly important that close collaboration or consultation with morphological experts is conducted and the stringent comparison of the specimens to be studied with the type specimen from the herbarium is carried out before commencing the molecular investigation. The future generation of taxonomists could apply more molecular genetic findings and should be encouraged to collaborate with molecular scientists to produce more solid and easy to use key and species descriptions for *Halophila* taxonomy.

It is anticipated that the major aims of achieving a clear differentiation of species and the reduction of difficult to determine species may be accomplished in the future under close collaboration between molecular scientists and taxonomists.

## 4. Materials and Methods

*Halophila* specimens from more than forty herbaria (Appendix A) and numerous freshly collected *Halophila* specimens from Australia, Hawaii, Florida, Thailand, Singapore and Japan were studied. Up to 50% of herbarium specimens were imaged.

The specimens were examined using light microscopes (dissecting microscope or low magnification optical microscope) to measure rhizome internodes (length and diameter), petiole length, blade length and width, the number of cross veins, the distance between adjacent cross veins as well as the distance between the intramarginal vein and the blade margin. These measurements were made on the mature leaves (the third leaves from the shoot apex). Numbers of cross veins were counted including those branches (see Figure 2). 

Some of the *Halophila* species which belong to sections other than section Halophila possess certain associated features such as hairs and seta on the blade surface and were required to be examined using a Philips Scanning Electron Microscope (SEM) operated at 15 kV to show their special 3D structures. Selected samples of seed surfaces were also imaged using the Scanning Electron Microscope. 

Reproductive structures such as flowers and fruits either from freshly collected or herbarium specimens were measured and imaged with light microscopes when they were available.

Botanical drawings of each *Halophila* species were made either from herbarium sheets or the freshly collected samples. The drawings of some species were made from more than one specimen. Sources of drawing samples for each species are listed in the Appendix A.

## Figures and Tables

**Figure 1 plants-09-01732-f001:**
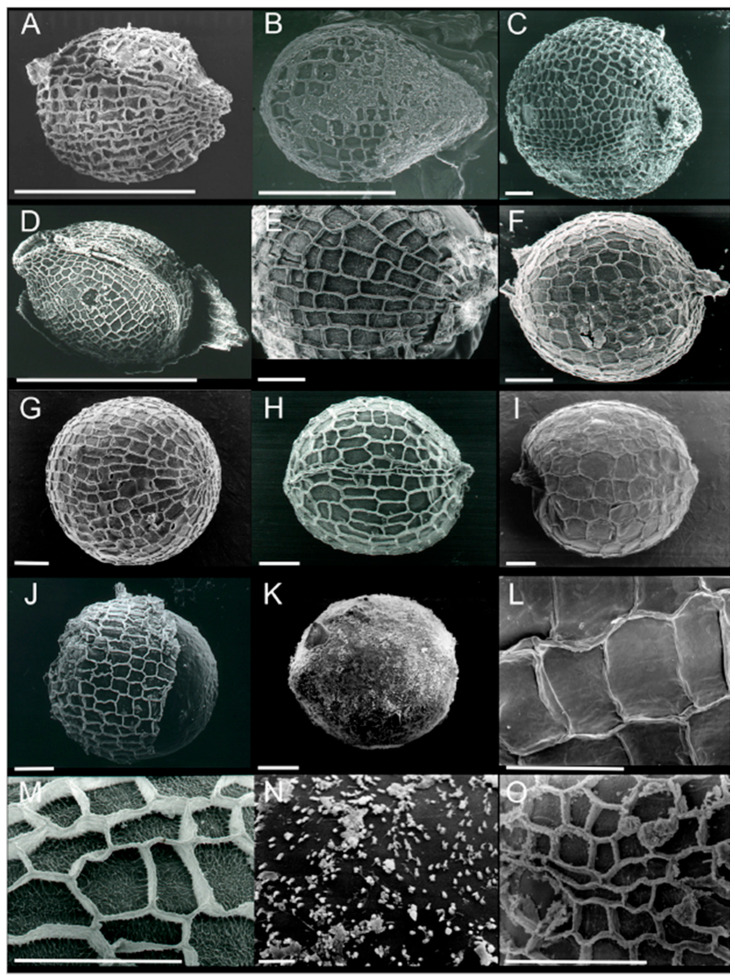
SEM Micrographs of seed surface in selected *Halophila* species (**A**): *H. australis*: SW Australia (UWA), scale bar = 1 mm., (**B**): *H. ovalis*: SW Australia (UWA), scale bar = 1 mm., (**C**): *H. ovalis*: Sri Lanka, *Meijer 782* (L), scale bar = 0.1 mm., (**D**): *H. madagascariensis*: Madagascar, *Bathie 14236* (P), scale bar = 1 mm., (**E**): *H. stipulacea*: Elate Gulf, *Lipkin 10174*, scale bar = 0.1 mm., (**F**): *H. decipiens*: Jamaica, *Throne 48288* (NY), scale bar = 0.1 mm., (**G**): *H. decipiens*: SW Australia (UWA), scale bar = 0.1 mm., (**H**): *H. decipiens*: Okinawa, Japan(UWA), scale bar = 0.1 mm., (**I**): *H*. sp. All. *decipiens*. NE Australia (W Lee Long), scale bar = 0.1 mm., (**J**): *H. decipiens*: Martinique, *Hahn 1271* (P), scale bar = 0.1 mm., (**K**): *H. tricostata*: Hinchinbrook Is. (QFH), scale bar = 0.1 mm., (**L**): *H.* all. *decipiens* sp.: NE Australia (QFH), scale bar = 0.1 mm., (**M**): *H. decipiens*: Proto Rico, *Howe 7039* (NY), scale bar = 0.1 mm., (**N**): *H. tricostata*: Queensland, Hinchinbrook Is. (QFH), scale bar = 0.1 mm., (**O**): *H. gaudichardii*: Guam, *Tsuda 3198* (GU), scale bar = 0.1 mm.

**Figure 2 plants-09-01732-f002:**
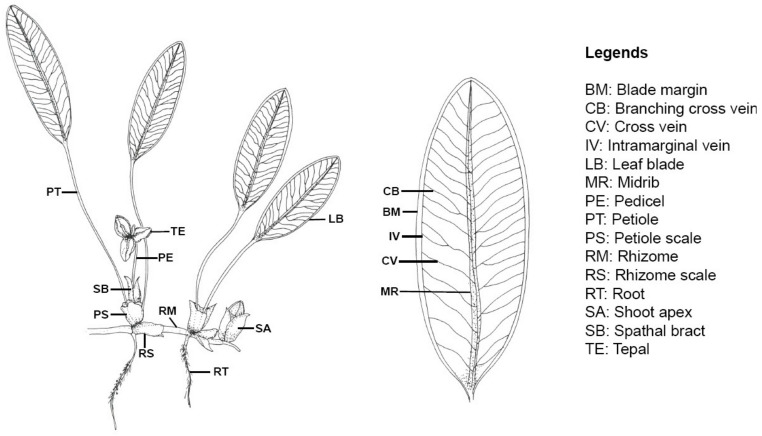
Botanical drawings of *Halophila* characteristics described in the keys and species descriptions.

**Table 1 plants-09-01732-t001:** A Brief History For The *Halophila* Taxonomy.

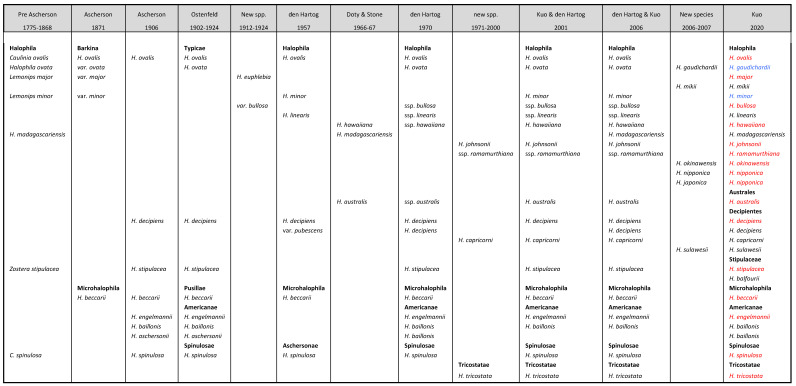

**Legend:** Vertical columns are years, periods, major researchers; horizontal columns are taxonomic species. **Bold** indicates Sections. *italic* indicates taxonomic species; var. represent varieties, all for *H. ovalis* varieties except *H*. *d**ecipiens* var. *pubescen*s. All ssp. indicates all subspecies belong to *H. ovalis*. Red colour indicates the species had molecular studies; blue colour indicates for the species identification the molecular studies were doubtful.

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
