# Peer review of "Taxonomy of the Genus Halophila Thouars (Hydocharitaceae): A Review"

_plants, 2020, doi:10.3390/plants9121732_

Round 1

Reviewer 1 Report

The review entitled "Taxonomy of The Genus Halophila Thouars (Hydocharitaceae): A Review " by Kuo is well written and an in depth review of genus Halophila. I was really enjoy reading this review and  the information summarized by this review will absolutely benefit to field scientist since some of the species in the genus Halophila are difficult to tell them apart in the field and without a comprehensive review yet. With the detail information on the morphological measurements presented in this study and those recommendations suggested by the author while dealing with species identification or evolution of this genus in the future are extremely helpful. The following are some of my comments on this review:

Line 470: The citation here should be "Ohba and Miyata (2007)"

Line 533: Something is missing here.

Line 824: It will be great if the author could include the SEM imagines of the epidermis mentioned in the manuscript since the author suggested microscopic approach is essential to some species in section Halophila. Therefore, these information could help to emphasis the importance of this methodology on species identification in this genus.

Line 832: "little known"

Line 881:

AFLP is not a next generation sequencing based method. There are few methods mentioned in Yu et al. 2018 (https://www.ncbi.nlm.nih.gov/pmc/articles/PMC6137260/) that you can consider to replace AFLP here.

Line 922: replace "generation sequence" by "sequencing"

In the appendix D: H. decipiens is not italic. 

Author Response

The author deeply appreciates the Reviewer’s positive and supportive comments on the manuscript and would like to thank him/her for providing the updated and appropriate reference regarding the newly developed sequencing methods.

Line 470: The citation here should be "Ohba and Miyata (2007)"   -corrected

Line 533: Something is missing here.    - corrected

Line 824: It will be great if the author could include the SEM imagines of the epidermis mentioned in the manuscript since the author suggested microscopic approach is essential to some species in section Halophila. Therefore, these information could help to emphasis the importance of this methodology on species identification in this genus.

SEM is mainly to observe the 3D objects, unfortunately, the leaf surfaces on the species in the Section Halophila rather flat and the epidermal cells of difference species have uniformly squamous appearance. Thus SEM is not an appropriate tool to be used to differentiate any structural differences and should not applied for species identification in the Section Halophila.

The author suggested using the light microscopy (dissecting and/or low magnification of optical microscope) to study the special features of different species for species identification purpose.

Line 832: "little known"   corrected

Line 881: corrected

AFLP is not a next generation sequencing based method. There are few methods mentioned in Yu et al. 2018 (https://www.ncbi.nlm.nih.gov/pmc/articles/PMC6137260/) that you can consider to replace AFLP here.

Line 922: replace "generation sequence" by "sequencing" -corrected

In the appendix D: H. decipiens is not italic. - corrected

Reviewer 2 Report

The author submitted a manuscript with the title "Taxonomy of The Genus Halophila Thouars (Hydocharitaceae): A Review".

The review gives a very broad and comprehensive overview about the current knowledge about the genus Halophila and includes important suggestions to solve the still open questions. The author describes all morphological traits in great details, included traits of reproductive organs and ecological aspects.

The high plasticity of members of the genus is not discussed. Consequently, even more sections are suggested whereas the reduction of sections and species can be deduced from a number of molecular marker studies. These points need to be at least included in the discussion.

Abstract

The abstract describes and summarizes the content of the review well.

Introduction

The first two paragraphs of the introduction could be written more general and less enumerative. The many changes in classification are not that interesting and typical for a number of taxa, but the current status is of more relevance.

Materials and Methods

This section could be more detailed. It need to be mentioned in detail for which drawings dried or fresh material was taken, respectively.

Other

The Appendix A is very helpful. The description of the supplementary material is not in agreement with the supplementary material and has to updated.

I would strongly recommend to include the very accurate, nice and important drawings into the main text of the paper to illustrate characteristics described in the keys.

Discussion

The discussion could be extended with respect to plasticity and therefore overlapping characters in populations of different species. Also the major aims of some taxonomist might be mentioned: clear differentiation of species and reduction of difficult to determine species.

Line 881: AFLP fingerprinting does not belong to new generation sequence methods. Please differentiate. More information about the time of separation of species might be interesting to involve and discuss.

Minor

Some writing errors need to be corrected (line 95 divide further and development, line 161, line 832, nown, line 894, line 904, earist somewhere in the text and some more).

What does "submitted" mean in line 828. Has there a manuscript been submitted? It might be better mentioned as unpublished data

Author Response

The author appreciates the Reviewer’s very constructive comments for improving the manuscript and agrees that “plasticity” may represent as a major concern in Halophila taxonomy. Indeed, prior to the development of molecular genetic research terms such as ‘plasticity’, ‘polymorphisms’ or ‘H. ovalis-H. minor species complex’ have often appeared in the literature indicating the extreme complexity and difficulty of defining the seagrass species particularly related to the Section Halophila. Based on the Reviewer’s valuable suggestions, this important issue will be addressed in the Discussion.

Reply to Introduction

For the biological taxonomy, the classification is very important and has to follow the very restricted rules set by “ICN” codes. The history of Halophila taxonomy may appear to be complicated and of little interest for readers outside the taxonomic field. However, it is essential for taxonomists to follow the history of genus/species name changes in order to use the correct names in their studies. For example, H. euphlebia was identified molecularly as an independent species from Japan in 2005. But the history indicated that H. euphlebia was the synonym of H. major. It is hoped that Table 1 could ‘reduce’ the complicated history of Halophila taxonomical development.

Sections (= subgenera) are purely for taxonomic use. If a genus has a large number of species, taxonomists will find additional classification characters to create another level of classification. Prior to this review, four out of five sections contain only one or two species each, while the section Halophila has 19 species. Among these 19 species, they can be further separated using reproductive and other morphological characters to reduce the number in the Section Halophila. Consequently, the three new designated sections all have only one to three species.

            This review presents all formally described Halophila species, including those rarely appearing in the literature such as H. madagascariensis, H. linearis, H. balfourii, etc. It is hoped that future molecular studies may exclude some of the described species to reduce the species number in the Section Halophila.

Materials and Methods (This session has been expanded as below:)

4.1. Halophila specimens from more than forty herbaria (Appendix B) and numerous freshly collected Halophila specimens from Australia, Hawaii, Florida, Thailand, Singapore and Japan were studied. Up to 50 % of herbarium specimens were imaged.

4.2. The specimens were examined using light microscopes (dissecting microscope or low magnification optical microscope) to measure rhizome internodes (length and diameter), petiole length, blade length and width, the number of cross veins, the distance between adjacent cross veins as well as the distance between the intramarginal vein and the blade margin. These measurements were made on the mature leaves (the third leaves from the shoot apex). Numbers of cross veins were counted including those branches.

4.3. Some of Halophila species which belong to sections other than section Halophila possess certain associated features such as hairs and seta on the blade surface and required to be examined using a Philips Scanning Electron Microscope (SEM) operated at 15 kV to show their special 3D structures.

4.4. Reproductive structures such as flowers and fruits either from freshly collected or herbarium specimens were measured and imaged with light microscopes when they were available.

4.5. Selected samples of seed surfaces were imaged using a Philips Scanning Electron Microscope at 15 kV. Sources of these specimen samples (Appendix D) were listed in the legend.

4.6. Botanical drawings of each Halophila species were made either from herbarium sheets or the freshly collected samples. The drawings of some species were made from more than one specimen. Sources of drawing samples for each species were listed in the legend (Appendix C).

Other

The Appendix A is very helpful. The description of the supplementary material is not in agreement with the supplementary material and has to updated.

Reply: The legend of the Appendix A has been corrected and updated.

I would strongly recommend to include the very accurate, nice and important drawings into the main text of the paper to illustrate characteristics described in the keys.

Reply: With your suggestion, I have prepared an additional and labelled botanical drawing (see attached). Hopefully, the Journal Editor will include in the main text.

Discussion

The discussion could be extended with respect to plasticity and therefore overlapping characters in populations of different species. Also the major aims of some taxonomist might be mentioned: clear differentiation of species and reduction of difficult to determine species.

Reply: I have modified some sections in Discussion to address these important messages.

Minor

Some writing errors need to be corrected (line 95 divide further and development, line 161, line 832, nown, line 894, line 904, earist somewhere in the text and some more).

Reply All have been corrected.

What does "submitted" mean in line 828. Has there a manuscript been submitted? It might be better mentioned as unpublished data

Reply: Yes, the paper has been submitted and is in press now.
